# Dynamic Response of Rock-like Materials Based on SHPB Pulse Waveform Characteristics

**DOI:** 10.3390/ma15010210

**Published:** 2021-12-28

**Authors:** Bi Sun, Rui Chen, Yang Ping, Zhende Zhu, Nan Wu, Yanxin He

**Affiliations:** 1Harbin Institute of Technology (Shenzhen), Shenzhen 518055, China; sunbi58@126.com (B.S.); cechenrui@hit.edu.cn (R.C.); 2Shenzhen Water Planning and Design Institute Co., Ltd., Shenzhen 518001, China; 3PowerChina Eco-Environment Group Co., Ltd., Shenzhen 518102, China; pingy@swpdi.com; 4Key Laboratory of Ministry of Education of Geomechanics and Embankment Engineering, Hohai University, Nanjing 210098, China; zhendezhunj@163.com; 5Guangzhou University-Tamkang University Joint Research Centre for Engineering Structure Disaster Prevention and Control, Guangzhou University, Guangzhou 510006, China; wunan@gzhu.edu.cn

**Keywords:** SHPB, waveform characteristics, strain rate, strain rate gradient, particle flow code

## Abstract

Rock-like brittle materials under dynamic load will show more complex dynamic mechanical properties than those under static load. The relationship between pulse waveform characteristics and strain rate effect and inertia effect is rarely discussed in the split-Hopkinson pressure bar (SHPB) numerical simulation research. In response to this problem, this paper discusses the effects of different pulse types and pulse waveforms on the incident waveform and dynamic response characteristics of specimens based on particle flow code (PFC). The research identifies a critical interval of rock dynamic strength, where the dynamic strength of the specimen is independent of the strain rate but increases with the amplitude of the incident stress wave. When the critical interval is exceeded, the dynamic strength is determined by the strain rate and strain rate gradient. The strain rate of the specimen is only related to the slope of the incident stress wave and is independent of its amplitude. It is also determined that the inertia effect cannot be eliminated in the SHPB. The slope of the velocity pulse waveform determines the strain rate of the specimen, the slope of the force pulse waveform determines the strain rate gradient of the specimen, and the upper bottom time determines the strain rate of the specimen. It provides a reference for SHPB numerical simulation. A dynamic strength prediction model of rock-like materials is then proposed, which considers the effects of strain rate and strain rate gradient.

## 1. Introduction

The split-Hopkinson pressure bar (SHPB) is an important piece of testing equipment for analyzing the dynamic mechanical properties of materials and is widely used in dynamic mechanical response analysis of rock-like brittle materials [1,2]. Rock-like brittle materials under dynamic load show more complex dynamic mechanical properties than those under static load [3]. The scatter that characterizes the experimental data contributes to the uncertainty and difficulty of interpreting such data [4]. Therefore, in order to obtain effective and accurate test data, constant strain rate deformation of the specimen must be guaranteed in the SHPB test [5,6].

Because the failure strain of rock-like brittle materials is very low, they often fail without reaching a constant strain rate under the condition of high-speed impact. In order to study the dynamic characteristics of rock-like brittle materials, many scholars first analyzed whether the incident wave waveform met the constant strain rate deformation of the specimen [7,8]. In early SHPB test research, the fatal effect of strain rate on the test results was not recognized [9,10]. It was determined that a pulse shaper could delay the rise time of the incident wave, and its appearance effectively improved the waveform [11,12,13]. Luo et al. [14] used an H62 brass sheet with a thickness of 2.5 mm and a diameter ranging from 3 to 10 mm as a pulse shaper to obtain different waveforms to achieve constant strain rate loading. Jiang et al. [15] obtained different dynamic strain rate-strain curves by different air gun pressures. Bagher et al. [16] studied the effects of striker bar velocity, diameter, and thickness of the pulse shaper on shaping pulses by SHPB tests experimentally and numerically. The shaping effect of hyperelastic alloy [17], copper material, or other soft materials [18,19] was also found to meet the constant strain rate deformation of rock. The incident waveform could also be artificially controlled by a variable cross-section incident bar [20] or special shape striker [21]. The results of numerical simulations showed that soft materials had more advantages in modifying the waveform [7,22]. Zhou et al. [23] carried out SHPB numerical simulation experiment by applying different impact speeds directly on the special-shaped striker. Kucewicz et al. [24] used a pulse shaper with a radius of 8 mm to conduct an SHPB experiment by LS-DYNA. Jankowiak et al. [25] used a cylindrical copper shaper with 20 mm diameter and 1 mm thickness. Baranowsk et al. [26] used three numerical modeling methods to establish the pulse shaper model to study the influence of mesh (particle) sensitivity on the characteristics of incident pulses.

The dynamic increase factor (DIF) can describe the variation of dynamic strength with strain rate. The dynamic strength of rock material DIF was found to have an exponential relation with the strain rate [27,28,29]. However, the test data often showed great discreteness [30,31] because the inertia effect was ignored. The inertia effect cannot be eliminated during dynamic loading [32] and is also a part of the macro-bearing capacity. While it is known that the inertia effect is caused by the rapid increase of specimens strain rate in a short time, there is no unified understanding of it at present. Rossi and other scholars [33,34] believed that the inertia effect became dominant under high strain rate; Guo et al. [35] determined that the inertia effect was significant when the strain rate was greater than 100/s; Zhou et al. [36] believed that the increase of material dynamic strength was due to the inertia effect rather than the strain rate effect; other scholars also put forward the dynamic strength criterion that considered the inertia effect to describe the dynamic strength of rock materials [37,38,39]. Li et al. [40] believed that the enhancement of the apparent dynamic strength of concrete materials is caused by the lateral inertial constraint rather than the strain rate sensitivity of the tested materials. Katayama et al. [41] considered if the mass is retained, then conservation of inertia and spatial continuity of inertia are maintained. The quadratic equation is used to describe the relation between DIF and log_10_ of the strain rate. Hao et al. [42,43,44] also used the quadratic equation to express the DIF model after considering lateral inertia. Al-Salloum [45] used the power function to express the DIF model. Lee et al. [46,47] proposed a new concrete DIF that excludes inertia effects by considering the strain acceleration and geometry of the specimens based on SHPB test results.

Numerical simulation is an important means to study the SHPB test [23]. Based on the particle flow code 2D (PFC2D) framework, this paper discusses the effects of different pulse types and waveforms on the specimen’s incident waveform and dynamic response characteristics. On the basis of previous theoretical analysis and numerical calculation results, a dynamic strength prediction model of rock-like brittle material is then proposed. This model comprehensively considers the effects of strain rate and strain rate gradient on macro dynamic strength.

## 2. Micro Model

The PFC2D framework was selected to simulate our SHPB impact experiment as it provides both a scientific tool to investigate the micromechanisms that combine to produce complex macroscopic behaviors and an engineering tool to predict these macroscopic behaviors [48]. PFC2D can be used to simulate many scientific problems in rock mechanics and rock engineering [49], and it is also widely used in SHPB dynamic testing [23,50,51,52,53].

### 2.1. Contact Bond Model

The PFC2D bonding model consists of a material system composed of grains and cement materials to jointly describe the micromechanical behavior of rock materials [54]. Grains and cement materials can be simplified into a spring and are equivalent to two microbeams, respectively, as shown in Figure 1. The framework includes two basic bonding models: parallel bond model (PBM) and contact bond model (CBM). The PBM provides the force-displacement behavior of a finite-sized piece of cementitious material deposited between two pieces, acting in parallel with a linear model [53]. The particle can freely move in the shear and normal direction and rotate between particles. The CBM is a simplification of the PBM [54]; that is, when the radius $\bar{R}$ (see Figure 1b) of the microbeam composed of the cement material is 0, the bond contact degenerates to one point. We employ the CBM for numerical simulation in this work. Although the microbeam of CBM cannot resist bending moment, it still has bond strength.

The expression of contact force $F_{c}$ in the CBM is [48]:(1)$$F_{c}=F_{\eta }+F_{l}$$
(2)$$F_{l}=k\delta$$
(3)$$F_{\eta }=2\beta \sqrt{mk}\dot{\delta }$$
where $F_{l}$ is linear force, $F_{\eta }$ is the damping force, β is the damping ratio, m is the particle mass, k is the contact stiffness, and $\delta ,\dot{\delta }$ are displacement and velocity, respectively.

The microbeam length $L=R_{A}+R_{B}$, $R_{A},R_{B}$ is the radius of particles A, B, respectively; The microbeam width $2\bar{R}$ is taken as the average value, i.e., $2\bar{R}=2\times \left(R_{A}+R_{B}\right)/2=R_{A}+R_{B}=L$; The microbeam thickness (particle disc thickness) is h (h=1 in PFC2D). Therefore, the particle contact area $A_{c}=Lh=L$. After simultaneously calculating Equations (1)–(3), and dividing by, divide by $A_{c}$, we obtain
(4)$$\sigma_{c}^{\prime }=k\varepsilon_{c}^{\prime }+\eta_{c}{\dot{\varepsilon }}_{c}^{\prime }$$
where $\sigma_{c}^{\prime }$ is the micro-contact stress, $\varepsilon_{c}^{\prime }$ is micro-strain, ${\dot{\varepsilon }}_{c}^{\prime }$ is the micro-strain rate, $\eta_{c}=2\beta \sqrt{mk}$ is the micro-damping coefficient.

### 2.2. SHPB Model

The numerical SHPB model is established, and its size is shown in Figure 2. The contact bond strength of bars and striker is selected to be high enough to avoid damage during the impact test. To enhance the contact between the specimen and the bar, the boundary particles at the contact between the bar and the bullet and between the bar and the specimen are refined into four layers. The refined particles have a radius of 0.3 mm and are aligned regularly [23], as illustrated in Figure 2. The radius and position of randomly generated particles obey uniform distribution within a limited range. The linear contact model is used between the bar and the bullet and between the bar and the specimen to simulate the state of contact without a bond. Three measuring circles are set on the incident bar, and one measuring circle is set on the transmission bar. The radius of the measuring circles is 20 mm, and the centers of measuring circles 1, 2, and 3 are 1150, 950, and 75 mm away from the right end of the incident bar, respectively. Measuring circle 5 is 375 mm away from the left end of the transmitted bar. Another measuring circle, 4, with a radius of 12.5 mm, is set on the specimen. The employed micro-parameters refer to references [22,50,55], as shown in Table 1.

### 2.3. Waveform Test

In order to investigate the influence of waveform characteristics on the dynamic response of rock specimens, two pulse types are used: velocity pulse and force pulse. The loading curve is designed as a trapezoid to control the shape of the waveform. To express the characteristics of the waveform, the waveform is controlled by amplitude, rise time, and the duration of the trapezoidal upper bottom. After many simulation tests, it is found that the trapezoidal upper bottom time has little effect on the test results for the velocity pulse. When it is taken as 0, the loading curve is too sharp and is inconsistent with the actual waveform curve. On the premise of not affecting the calculation results and reducing the calculation time, the bottom time is set to 10 μs. As the rise time and bottom time of the pulse waveform significantly influence the force pulse calculation results, the pulse shape is adjusted according to the calculation requirements. The rise and fall times of the pulse curve are the same in the two pulse modes. Two typical pulse waveforms are shown in Figure 3.

Using the pulse waveform curve in Figure 3b, the stress waveform on the Hopkinson bars is obtained, as shown in Figure 4. In the figure, the stress waveforms measured by measuring circles 1, 2, and 3 on the incident bar are almost the same, indicating that the waveform has no obvious dispersion phenomenon. Additionally, the stress waveforms measured by measuring circles 1, 2, and 3 on the incident bar are almost the same, indicating that the incident wave has no obvious dispersion phenomenon. Thus, it can be determined that the established SHPB numerical model is basically reliable.

## 3. Velocity Pulse

### 3.1. Pulse Waveform

The shape of the velocity pulse waveform is controlled by changing amplitude $A_{v}$ and rise time $t_{r}$. We study the law of rock dynamic response in this paper. Five groups of velocity pulses with different amplitudes and the same slope of the rising section curve are adopted, as shown in Figure 5a. The slope of the five groups of pulse curves is 50,000 m/s^2^. The amplitude *A*_v_ = 8, 10, 12, 14 and 16 m/s, respectively. Five groups of velocity pulses with the same amplitude and different slopes of the rising section curve are adopted, in which the slope of the pulse curve is controlled by rise time $t_{r}$, as shown in Figure 5b. The rise time *t*_r_ = 50, 100, 150, 200 and 250 µs respectively, and *A*_v_ = 12 m/s.

### 3.2. Different Amplitudes

The numerical calculation results of different amplitudes are shown in Figure 6. The strain and strain rate ${\dot{\varepsilon }}_{c}$ of the specimen can be obtained by monitoring the displacement of the particles at the left and right ends of the specimen. The strain rate ${\dot{\varepsilon }}_{c}$ is obtained by the derivative of the strain $\varepsilon_{c}$ to time.

Figure 6a shows the time history curves of the stress monitored by measuring circle 1 on the incident bar. The following incident stress waves are taken from the monitoring data of measuring circle 1. The shape of the time history curve of stress is consistent with its velocity pulse waveform. The slope of the curve loading stage, i.e., the incident wave stress rate ${\dot{\sigma }}_{i}$, is the same, and ${\dot{\sigma }}_{i}=2450\;\mathrm{GPa}/s$.

Figure 6b shows the time history curve of the strain rate with the same slope but using the different amplitudes of the five groups of pulse curves. The curve overlap of the five groups is very high, indicating that the strain rate of the specimen remains unchanged when only changing the amplitude of the velocity pulse curve without changing the slope. The time history curve of strain rate in the loading stage is divided into three stages. The time history curve of the stress (the dotted line in Figure 6b at $A_{v}=12\;m/s$ is given to facilitate the division of the three stages. The first stage is the initial non-constant strain rate loading stage S_1_, in which the strain rate increases sharply, and the specimen particles are dominated by the inertial effect. Under the axial dynamic load, part of the work of the external load is to provide kinetic energy to the particles so that the particles obtain axial acceleration. Due to the existence of the Poisson effect, particles will interact with adjacent particles, and adjacent particles will obtain radial acceleration. The load that provides acceleration to the particles is the inertial effect, which is part of the macro-bearing capacity [37]. The inertia effect can be characterized by strain rate gradient ${\ddot{\varepsilon }}_{c}$ [56], which can be obtained by the derivative of strain rate ${\dot{\varepsilon }}_{c}$ to time t. The strain rate gradient of the specimen in S_1_ ${\ddot{\varepsilon }}_{c}=1.16\;/s^{2}$. The second stage is the constant strain rate loading stage S_2_, where the specimen is subjected to constant strain rate loading. The strain rates ${\dot{\varepsilon }}_{c}$ of the five groups of specimens are the same, ${\dot{\varepsilon }}_{c}=30.98\;/s$. In this stage, the specimen particles keep moving at a uniform speed, and the strength has an exponential relation with the strain rate. Only when the rock deformation is in the state of constant strain rate can the results can be used as the constitutive relation [10]. The third stage is the non-constant strain rate stage S_3_ of accelerated failure. At this time, the inner microcracks of the specimen gradually form macro cracks. The force on the specimen is unbalanced, and the inner particle motion intensifies. The acceleration caused by the interaction between particles accelerates the failure of the specimen. The inertia effect also plays a leading role in this stage. Since the inertia effect is a part of the macro-bearing capacity, the specimen’s macro bearing capacity also increases at this stage.

The stress–strain curve in Figure 6c shows that when the amplitude $A_{v}=8\;m/s$, the peak stress $\sigma_{p}=417.95\;\mathrm{MPa}$, and the peak stress of the other four groups $\sigma_{p}=442.86\;\mathrm{MPa}$. The stress of the specimen is taken from the monitoring data of measuring circle 4, the same below. Figure 6c also shows the distribution of specimen failure cracks at peak stress with $A_{v}=8\;m/s$ and one of the other four groups. The failure modes of the other four groups are exactly the same, and the number of microcracks is 2499. When $A_{v}=8\;m/s$, the number of microcracks in the specimen is 2260, and its failure mode is similar to that of the other four groups. It can be seen that compared with the failure mode when $A_{v}=8\;m/s$, the other four groups are further damaged. In the previous study of rock dynamics, the strain rate was the only variable used to characterize the dynamics, meaning that the peak stress should be the same with the same strain rate. This assumption is inconsistent with the calculation results in this paper. The possible reason for this phenomenon is that a specimen has a critical interval of dynamic strength. When the amplitude is less than this critical interval, the specimen will not be damaged; when the amplitude is in this critical interval, the specimen will be damaged. Its dynamic strength will increase with the amplitude, but it is independent of the strain rate. This is because when the amplitude increases, the inner particle interaction force of the specimen increases, but the internal crack expands further before it has time to penetrate. When the amplitude is greater than this critical interval, the dynamic strength will not increase no matter how the amplitude increases because the inner cracks of the specimen have penetrated, and the stress–strain curve remains unchanged. Therefore, when the strain rate is used to characterize the dynamic stress of the specimen, whether it is in the critical interval should also be considered.

### 3.3. Different Rise Times

The SHPB impact numerical simulation test is carried out on the specimen using five groups of pulse curves with different rise times, as shown in Figure 5b. The calculation results are provided in Figure 7.

Figure 7a shows the time history curves of the incident stress wave, where the curve shape is consistent with the velocity pulse waveform. The slopes of the five groups of stress wave curves are different, and their peak stresses $\sigma_{p}$ are similar, $\sigma_{p}\;\approx \;589\;\mathrm{MPa}$. Figure 7b,c show that with the amplitude unchanged, the strain rate and peak stress increase as the rise time of the pulse curve shortens. Compared with the analysis results in Section 3.2, the strain rate of the specimen is only related to the slope of the incident stress waveform (i.e., the incident wave stress rate, the derivative of stress to time) but independent of its amplitude. As illustrated in Figure 7b, when the curve rise time $t_{r}=250\;\mu s$, the specimen can also maintain the constant strain rate loading. However, when the rise time is shortened to 50 μs, the specimen cannot maintain constant strain rate loading. Figure 7c shows the failure crack distributions of the specimen at the peak stress of the five groups of curves. It can be seen that the failure modes of the five groups are similar. The shorter the rise time, the greater the crushing degree of the specimen at the peak stress.

As shown in Figure 8a, the fitting degree between the incident wave stress rate ${\dot{\sigma }}_{i}$ and specimen strain rate ${\dot{\varepsilon }}_{c}$ is very high, $R^{2}=0.9997$.

The fitting degree is determined according to Equation (5):(5)$${\dot{\sigma }}_{i}=0.084{\dot{\varepsilon }}_{c}-0.21,\;R^{2}=0.9997$$

The strain rate of the specimen has a logarithmic relationship with the peak stress [28,52]. It is also possible to characterize the peak stress by the incident wave stress rate. In Figure 8b, the incident wave stress rate and strain rate of the specimen are used to fit the peak stress, and their fitting curves and degrees are very similar, as shown in Equation (6). Therefore, we believe that the strain rate of the specimen is only related to the slope of the incident stress waveform and is independent of its amplitude.
(6)$$\left\{ \begin{array}{l} \sigma_{p}=115.21\log_{10}({\dot{\sigma }}_{i})+414.13,\;R^{2}=0.8502\\ \sigma_{p}=119.25\log_{10}({\dot{\varepsilon }}_{c})+280.85,\;R^{2}=0.8585 \end{array}\right.$$

However, the fitting degree of the curve is not high, which may indicate that it is not appropriate to simply use strain rate or stress rate fitting and that the strain rate gradient should also be taken into account.

## 4. Force Pulse

### 4.1. Pulse Waveform

In the force pulse waveform, the numerical simulation test is carried out by changing the amplitude $A_{f}$, rise time $t_{r}$, and upper bottom time $t_{u}$ of the waveforms. Each case is divided into five groups, with a total of fifteen groups. In the five groups of waveforms with different amplitudes, the pulse slope $\dot{f}$ and upper bottom time $t_{u}$ remain unchanged, $\dot{f}=250\;N/s$ and $t_{u}\;=100\;\mu s$, respectively. The amplitudes *A*_f_ = 3.0, 3.5, 4.0, 4.5 amd 5.0 kN, respectively. In the five groups of waveforms with different rise times, the amplitude $A_{f}$ and upper bottom time $t_{u}$ remain unchanged, $A_{f}=4.0\;\mathrm{kN}$ and $t_{u}=100\;\mu s$, respectively. The rise time $t_{r}=20\;\mu s,\;30\;\mu s,\;40\;\mu s,\;50\;\mu s,\;60\;\mu s$, respectively. In the five groups of waveforms with different upper bottom, the amplitude $A_{f}$ and rise time $t_{r}$ remain unchanged, $A_{f}=4.0\;\mathrm{kN}$ and $t_{r}=40\;\mu s$, respectively. The upper bottom time t_u_ = 50, 75, 100, 150 and 200 µs, respectively. The pulse waveform is shown in Figure 9.

### 4.2. Different Amplitudes

The SHPB impact numerical simulation test is carried out on the specimen by using the amplitude of the five groups of different force pulse curves in Figure 9a. The calculation results are shown in Figure 10.

Figure 10a shows that the shape of the incident stress waveform is not consistent with that of the force pulse waveform. The amplitude and slope of the incident stress waveform increase with the amplitude of the pulse waveform amplitude. Since the rising slope of the five groups of pulse waveforms is the same, the incident stress waveforms almost completely overlap in the initial loading stage (marked in the figure). The strain rate gradient ${\ddot{\varepsilon }}_{c}$ in the initial non-constant strain rate loading stage S_1_ (marked when $A_{f}=3.0\;\mathrm{kN}$) in Figure 10b is also relatively similar. This indicates that the strain rate gradient ${\ddot{\varepsilon }}_{c}$ can be controlled by the rising slope of the force pulse waveform. When the upper bottom time of the pulse waveform acts continuously, the stress in the incident bar also increases gradually until the action time of the upper bottom ends. Therefore, the greater the amplitude of the pulse waveform, the greater the rate of stress increase in the incident bar, and the greater the slope and amplitude of the stress time history curve. According to the above analysis, the strain rate of the specimen in S_2_ is related to the rising slope of the incident wave. Therefore, the strain rate increases with the incident wave amplitude in Figure 10b.

Although a certain characteristic of the incident waveform cannot be controlled by using force pulse waveforms with different amplitudes, the analysis shows that the rising slope of the force pulse waveform determines the strain rate gradient of the specimen, and the upper bottom time determines the amplitude of the incident stress wave. The simulation test scheme with different rise and upper bottom times is designed on this basis.

### 4.3. Different Rise Time

The different strain rate gradients ${\ddot{\varepsilon }}_{c}$ can be obtained by changing the rise time while keeping the amplitude $A_{f}$ and the rise time $t_{r}$ of the force pulse waveform unchanged. However, when the rise time is too large, the rising slope of the incident waveform decreases, and the strain rate will also decrease. Thus, it is necessary to control the rise time to obtain different strain rate gradients at the same strain rate. According to our simulations, when $A_{f}=4\;\mathrm{kN}$ and $t_{r}\leq 60\;\mu s$, the strain rate time history curve with the same strain rate and different strain rate gradients can be obtained. The force pulse waveform is shown in Figure 9b. The calculation results are shown in Figure 11.

In Figure 11a, the duration of the initial loading stage of five incident stress waves increases with the rise time of the force pulse wave. The initial loading section when $t_{r}=60\;\mu s$ is marked in the figure. The rise slope ${\dot{\sigma }}_{i}$ and amplitude of the incident stress wave are the same, where ${\dot{\sigma }}_{i}=5200\;\mathrm{GPa}/s$. In Figure 11b, the strain rates of the five curves in S_2_ remain basically the same, ${\dot{\varepsilon }}_{c}=58.8\;/s$. The slope of the curve, i.e., the strain rate gradient ${\ddot{\varepsilon }}_{c}$, in S_1_ ($t_{r}=20\;\mu s$ marked in the figure), is significantly different, indicating that different strain rate gradients can be obtained by controlling the loading slope with different rise times of force pulse waveform. The smaller $t_{r}$ is, the greater ${\ddot{\varepsilon }}_{c}$ is. The peak stress of the five curves in Figure 11c increases with the decrease of the rise time. Therefore, the greater the strain rate gradient ${\ddot{\varepsilon }}_{c}$, that is, the greater the inertia effect, the greater the peak stress.

The strain rate gradient ${\ddot{\varepsilon }}_{c}$ can be obtained by extracting the slope of five curves in S_1_, Figure 11b. Since the strain rate gradient is caused by the acceleration between inner particles in the specimen, the peak stress $\sigma_{p}$ and strain rate gradient ${\ddot{\varepsilon }}_{c}$ are fitted by a linear function. The fitting curve is shown in Figure 12.

The fitting relation between peak stress $\sigma_{p}$ and strain rate gradient ${\ddot{\varepsilon }}_{c}$ is shown in Equation (7).
(7)$$\sigma_{p}=26.449{\ddot{\varepsilon }}_{c}+430.97,\;R^{2}=0.9939$$
where the fitting degree $R^{2}$ reaches 0.9939. The high fitting degree shows that there is a highly significant linear correlation between $\sigma_{p}$ and ${\ddot{\varepsilon }}_{c}$. When the rising slope of the force pulse waveform is large, the strength of the specimen is dominated by the inertia effect. Comparing the fitting degree between strain rate and peak strength in Equation (6), it can be determined that the inertia effect must not be ignored in SHPB.

### 4.4. Different Upper Bottom Times

The numerical simulation test of SHPB pulse is carried out on the specimen by using the upper bottom time of the five different force pulse waveforms in Figure 9c. The calculation results are shown in Figure 13.

Figure 13a shows that the initial loading stage and rising stage of the five time history curves of incident stress waves are completely overlapped. Accordingly, in Figure 13b, the strain rate time history curves are overlapped in the S_1_ and S_2_ stages. Thus, the amplitude of the time history curve of the incident stress wave will increase with the upper bottom time of the force pulse waveform. When the upper bottom time $t_{u}\geq 100\;\mu s$, the amplitude does not increase. Figure 13c shows that when $t_{u}<75\;\mu s$, the peak stress of the specimen increases with the upper bottom time. When $t_{u}\geq 75\;\mu s$, the peak stress does not increase. This is the same as the law obtained by using velocity pulse waveforms with different amplitudes in Section 3.2. On the one hand, the impact effect of the upper bottom time of the force pulse waveform and the amplitude of the velocity pulse waveform are the same. On the other hand, a critical interval for the dynamic bearing strength of the specimen is also identified. The figure shows the crushing morphology of specimens at different peak stresses. The crushing morphology of specimens is relatively similar, and the number of microcracks increases with the increase of peak stress.

## 5. Dynamic Strength Prediction Model

According to the previous analysis, the strain rate and strain rate gradient of rock specimens will affect their dynamic strength. Here, they are regarded as independent variables, and a rock dynamic strength prediction model is established, where the strain rate ${\dot{\varepsilon }}_{c}$ is fitted by a logarithmic function, and the strain rate gradient ${\ddot{\varepsilon }}_{c}$ is fitted by a linear function. The prediction model is shown in Equation (8):(8)$$\sigma_{p}=a\log_{10}({\dot{\varepsilon }}_{c})+b{\ddot{\varepsilon }}_{c}+c$$
where *a*, *b*, and *c* are fitting values.

In the time history curves of strain rate obtained by using velocity pulse waveforms with different slopes in Section 3.3, the strain rates and strain rate gradients of the five curves are different. This prediction model is used to refit the curve in Figure 8, and a two-dimensional fitting curve is adopted to facilitate visual expression. As shown in Figure 14, $\left(a\ln({\dot{\varepsilon }}_{c})+b{\ddot{\varepsilon }}_{c}\right)$ is set as the abscissa, and the fitting relation is shown in Equation (9):
(9)$$\sigma_{p}=308.90\log_{10}({\dot{\varepsilon }}_{c})–28.98{\ddot{\varepsilon }}_{c}+21.85,\;R^{2}=0.9806$$

As illustrated in Figure 14, the fitting curve obtained using the prediction model in Equation (8) can effectively describe the dynamic peak strength of the specimen under the impact of the velocity pulse waveform with different slopes. The fitting degree of Equation (9) $R^{2}=0.9806$ is also greatly improved compared with that of Equation (6) $R^{2}=0.8585$.

The simulation test data under the force pulse waveform are selected to verify the prediction model. As mentioned in Section 4.3, in the case where the amplitude and upper bottom time of the force pulse waveform remain unchanged, when the rise time is too large, both the strain rate gradient and the strain rate will decrease, while the amplitude $A_{f}$ and upper bottom time $t_{u}$ remain unchanged. The rise time $t_{r}=50\;\mu s,\;100\;\mu s,150\;\mu s,\;200\;\mu s,\;300\;\mu s$, respectively. The calculation results are shown in Figure 15. 

Figure 15a,b show that when the amplitude and upper bottom time of the force pulse waveform remain unchanged but the rise time is too large, the strain rate gradient, strain rate, and peak strength will decrease with the increase of rise time. Figure 15c shows the fitting curve obtained by using the prediction model in Equation (8), and the fitting formula is as follows:(10)$$\sigma_{p}=132.20\log_{10}({\dot{\varepsilon }}_{c})+3.09{\ddot{\varepsilon }}_{c}+221.69,\;R^{2}=0.9771$$

The fitting degree $R^{2}$ reaches 0.9771, which indicates that the prediction model can effectively describe the dynamic peak strength of the specimen under the impact of a force pulse waveform with different rise times.

Lee’s test results [47] are used to verify the reliability of the model. Three groups of data with the same sample size but different static compressive strengths are extracted. The sample size is 50 mm × 50 mm. The fitting result is shown in Figure 16, and the fitting formula is shown in Equation (11).
(11)$$\left\{ \begin{array}{l} \sigma_{PS52}=73.75\log_{10}({\dot{\varepsilon }}_{c})–4.4763{\ddot{\varepsilon }}_{c}–71.31,\;R^{2}=0.6934\\ \sigma_{PS61}=35.65\log_{10}({\dot{\varepsilon }}_{c})+3.5120{\ddot{\varepsilon }}_{c}+10.5,\;R^{2}=0.9662\\ \sigma_{PS85}=–9.15\log_{10}({\dot{\varepsilon }}_{c})+17.62{\ddot{\varepsilon }}_{c}+106.80,\;R^{2}=0.8987 \end{array}\right.$$
where $\sigma_{PS52}$, $\sigma_{PS61}$, $\sigma_{PS85}$ represent the dynamic strength with static compressive strength of 51.90 MPa, 61.35 MPa, and 85.10 MPa, respectively. The fitting degrees $R^{2}$ are 0.6934, 0.9662, and 0.8987, respectively. It shows that the model proposed in this paper can accurately describe the dynamic strength of the specimen.

## 6. Discussion

The slope of the loading curve of the velocity pulse is acceleration, and the force is generated when the acceleration acts on the mass particles. If the slope remains constant, the particles will be subjected to a constant force. Comparing the loading modes of the two velocity pulses, it can be found that when loading with different amplitudes, the slope of each pulse curve is the same, the constant force and motion state of the particles are the same on the meso-scale, and the mechanical properties of the specimens are the same on the macro scale. However, when loading with different rise times, the slope of the pulse curve is different, and the constant force on the particles is also different. The greater the constant force, the greater the force between particles, and the greater the intensity of particle motion. Macroscopically, it shows the increase of strain rate and strength of the specimen.

The slope of the loading curve of the force pulse is a jerk, which is the change rate of acceleration with time. The shorter the rise time of the curve, the greater the slope, the faster the acceleration rate increases. The force between particles is increasing rapidly, and the strain rate gradient and inertia effect of the specimen are also increasing. When the amplitude and slope of the curve are kept constant, the force between particles gradually increases to the set value over time and remains constant. At this time, the length of the upper bottom determines the duration of the constant force, but it does not change the strain rate and strain rate gradient of the specimen. Therefore, the velocity pulse directly affects the movement rate of particles in the specimen, which determines the strain rate of the specimen. The force pulse directly affects the acceleration generated by particle interaction, which determines the strain rate gradient of the specimen.

By comparing the incident waveforms obtained by two loading types of velocity and force pulse, it is found that the triangular waveform of the incident waveform is similar to its velocity pulse, while the incident wave waveform obtained by the trapezoidal shape of the force pulse is similar to the sine waveform. In the “initial loading stage” of the sine waveform, there is acceleration between particles, and the mechanical properties of the specimen are determined by the acceleration (i.e., inertia effect). The nonlinear characteristic can be used to control the strain rate gradient of the specimen. According to this character, the strain rate gradient can be fixed and the strain rate can be changed, as shown in Figure 10b, or the strain rate can be fixed and the strain rate gradient can be changed, as shown in Figure 11b. However, a triangular wave can not fix one parameter and change the other at the same time. Different slopes will result in different strain rates and strain rate gradients, as shown in Figure 7b. Because there is no nonlinear characteristic of “initial loading section” in a triangular wave, the strain rate effect and inertia effect of the specimen are effectively controlled. Therefore, the trapezoidal shape of force pulse is more suitable for SHPB research.

## 7. Conclusions

A numerical simulation of rock SHPB was carried out in this work, based on the characteristics of the pulse waveform. The effects of different pulse types and pulse waveforms on the dynamic response characteristics of specimens were discussed. The following conclusions were obtained:

1.There is a critical interval of rock dynamic strength. Below the critical interval, the specimen will not be damaged. In the critical interval, the dynamic strength of the specimen is independent of the strain rate but increases with the amplitude of the incident stress wave. When the critical interval is exceeded, the dynamic strength is determined by strain rate and strain rate gradient.2.The strain rate of the specimen is only related to the slope of the incident stress wave and is independent of its amplitude. Additionally, the inertia effect cannot be eliminated in SHPB.3.The slope of the velocity pulse waveform determines the strain rate of the specimen. The slope of the force pulse waveform determines the strain rate gradient of the specimen, and the upper bottom time determines the strain rate of the specimen. The two pulse types can achieve the same impact effect by controlling the waveform shape.4.A dynamic strength prediction model of rock-like brittle materials is proposed. The model considers the effects of strain rate and strain rate gradient and is verified.

## Figures and Tables

**Figure 1 materials-15-00210-f001:**
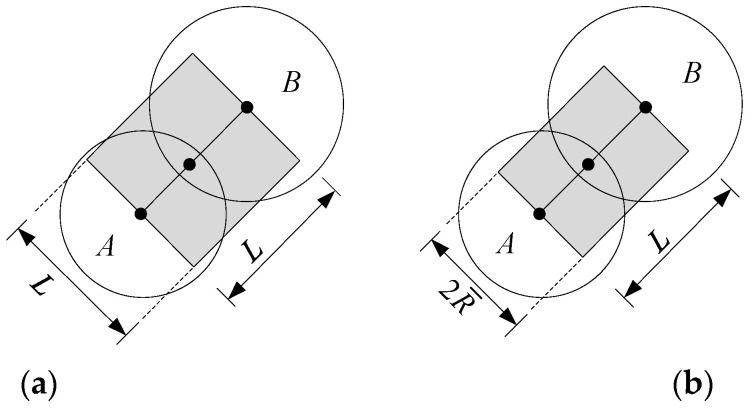
Equivalent meso-beam model of the grain-cement system [54]. (**a**) Grain behavior; (**b**) cement behavior.

**Figure 2 materials-15-00210-f002:**
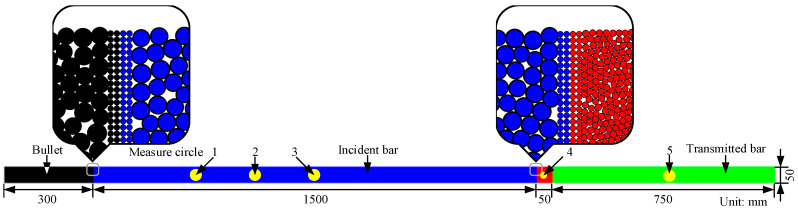
SHPB model.

**Figure 3 materials-15-00210-f003:**
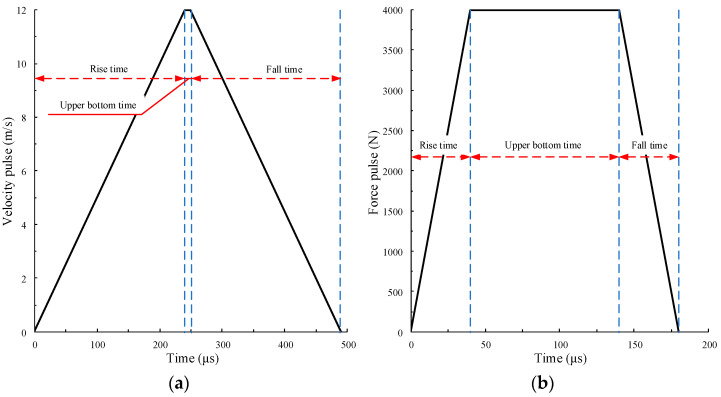
Typical pulse waveforms. (**a**) Velocity pulse; (**b**) force pulse.

**Figure 4 materials-15-00210-f004:**
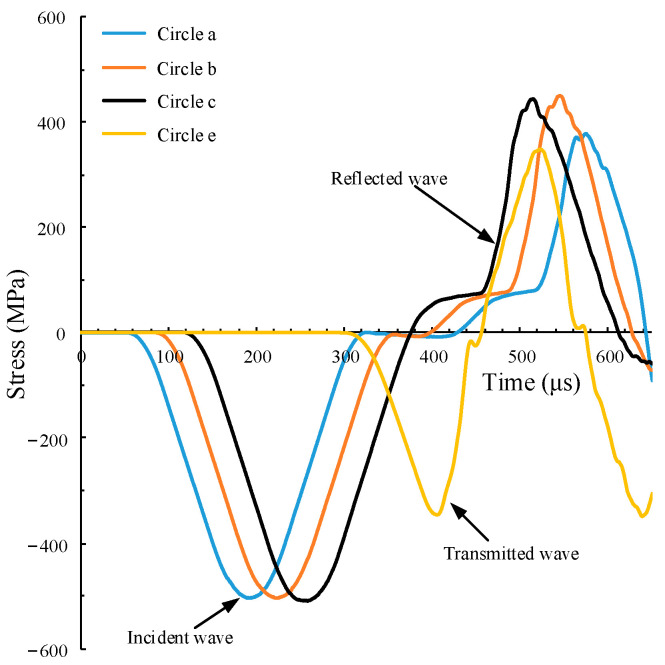
Stress waves recorded on the Hopkinson bars.

**Figure 5 materials-15-00210-f005:**
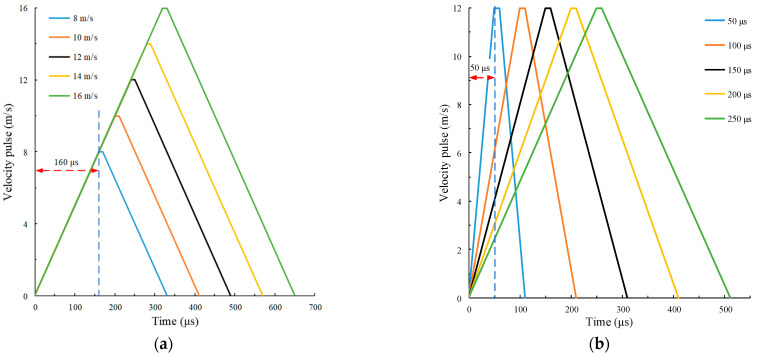
Velocity pulse waveform. (**a**) Different amplitude; (**b**) different rise time.

**Figure 6 materials-15-00210-f006:**
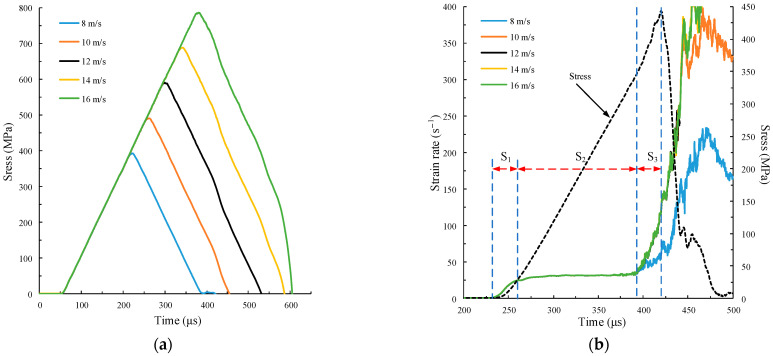
Calculation curve under different amplitudes. (**a**) Time history curves of incident stress wave; (**b**) time history curves of strain rate and stress; (**c**) stress–strain curve.

**Figure 7 materials-15-00210-f007:**
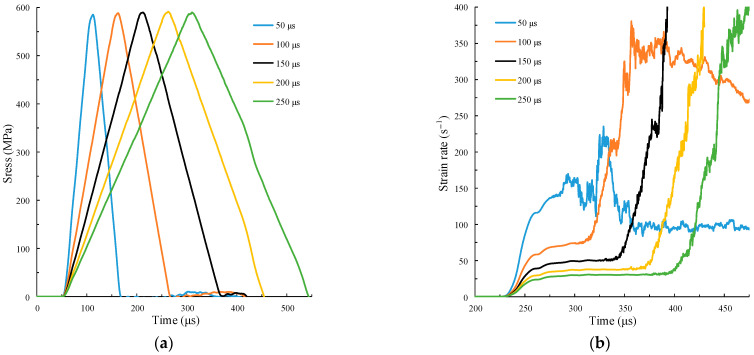
Curve calculations under different slopes. (**a**) Time history curves of incident stress wave; (**b**) time history curves of strain rate; (**c**) stress–strain curve.

**Figure 8 materials-15-00210-f008:**
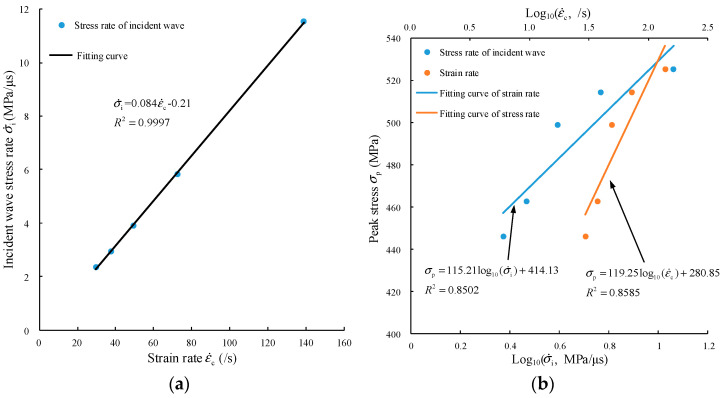
Fitting relationship between the incident wave stress rate and strain rate and peak stress. (**a**) Relationship between stress rate and strain rate; (**b**) relationship between peak stress and stress rate and strain rate.

**Figure 9 materials-15-00210-f009:**
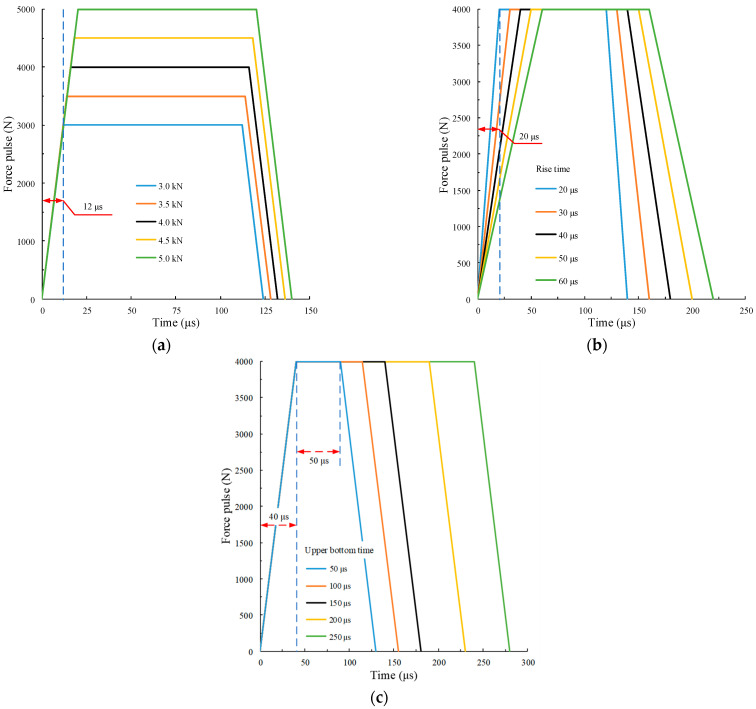
Velocity pulse waveform. (**a**) Different amplitudes; (**b**) different rise times; (**c**) different upper bottom times.

**Figure 10 materials-15-00210-f010:**
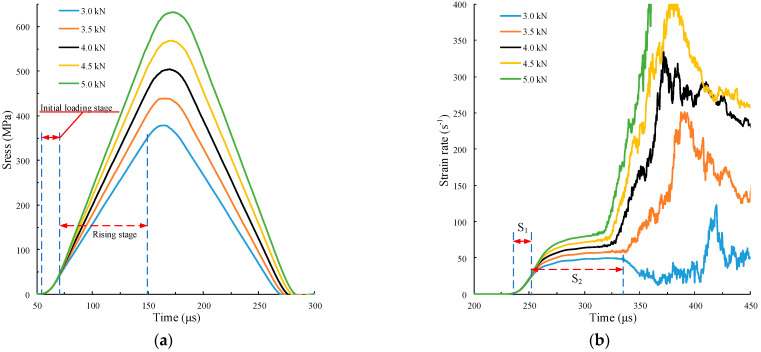
Calculation curve under different amplitudes. (**a**) Time history curves of incident stress wave; (**b**) time history curves of strain rate.

**Figure 11 materials-15-00210-f011:**
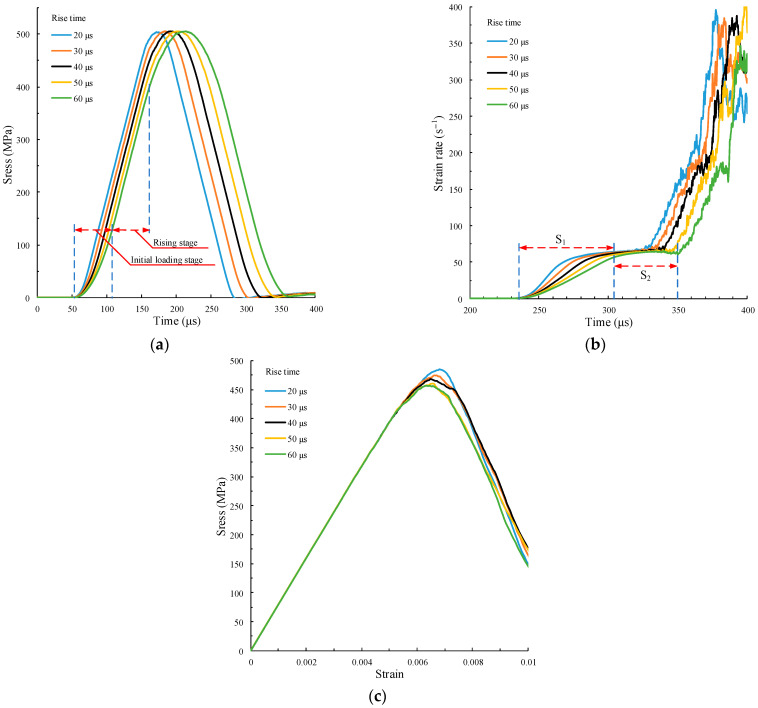
Curve calculations under different rise times. (**a**) Time history curves of incident stress wave; (**b**) time history curves of strain rate; (**c**) stress–strain curve.

**Figure 12 materials-15-00210-f012:**
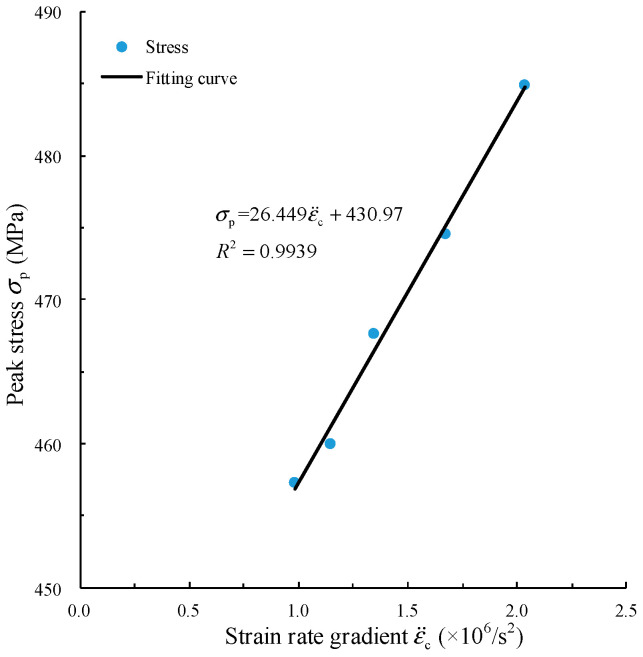
Fitting relation between peak stress and strain rate gradient.

**Figure 13 materials-15-00210-f013:**
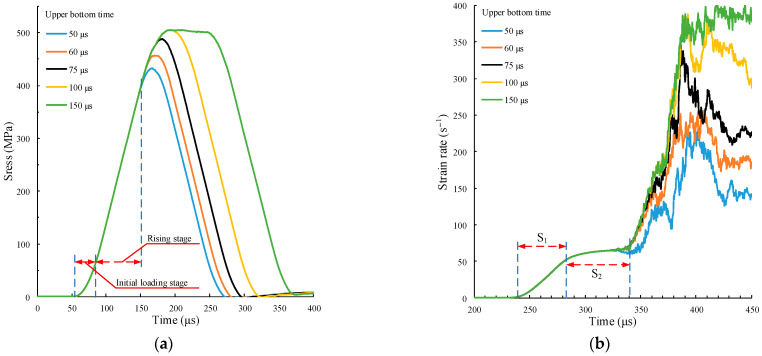
Curve calculations under different upper bottom times. (**a**) Time history curves of incident stress wave; (**b**) time history curves of strain rate; (**c**) stress–strain curve.

**Figure 14 materials-15-00210-f014:**
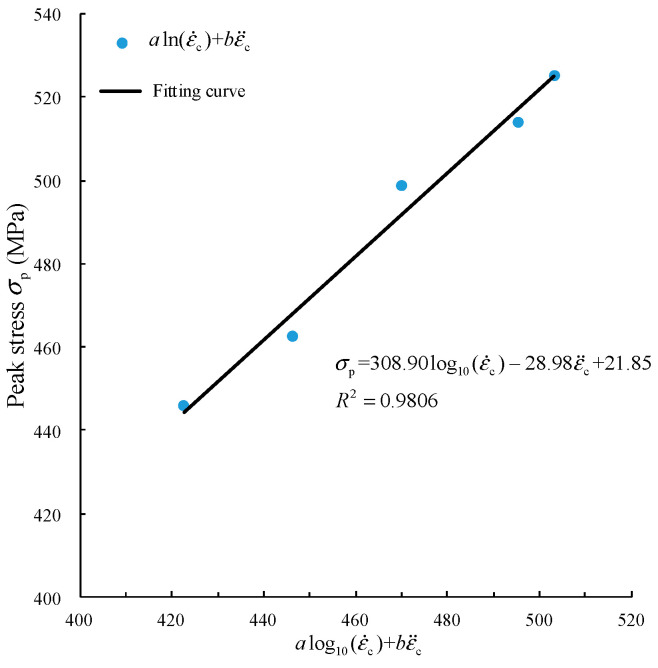
Fitting relationship between peak stress and strain rate and strain rate gradient.

**Figure 15 materials-15-00210-f015:**
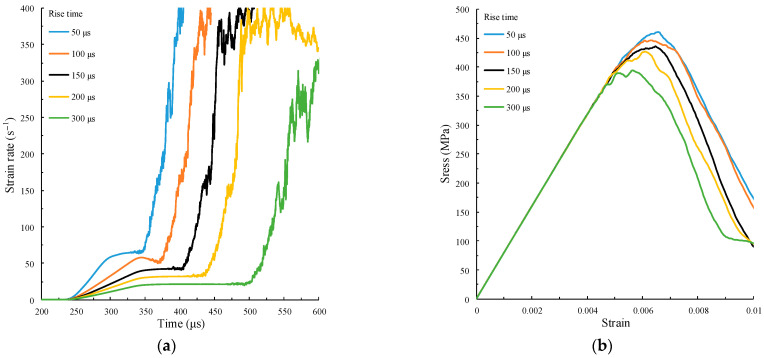
Calculation results under different rise times. (**a**) Time history curves of strain rate; (**b**) stress–strain curve; (**c**) fitting relation between peak stress and strain rate and strain rate gradient.

**Figure 16 materials-15-00210-f016:**
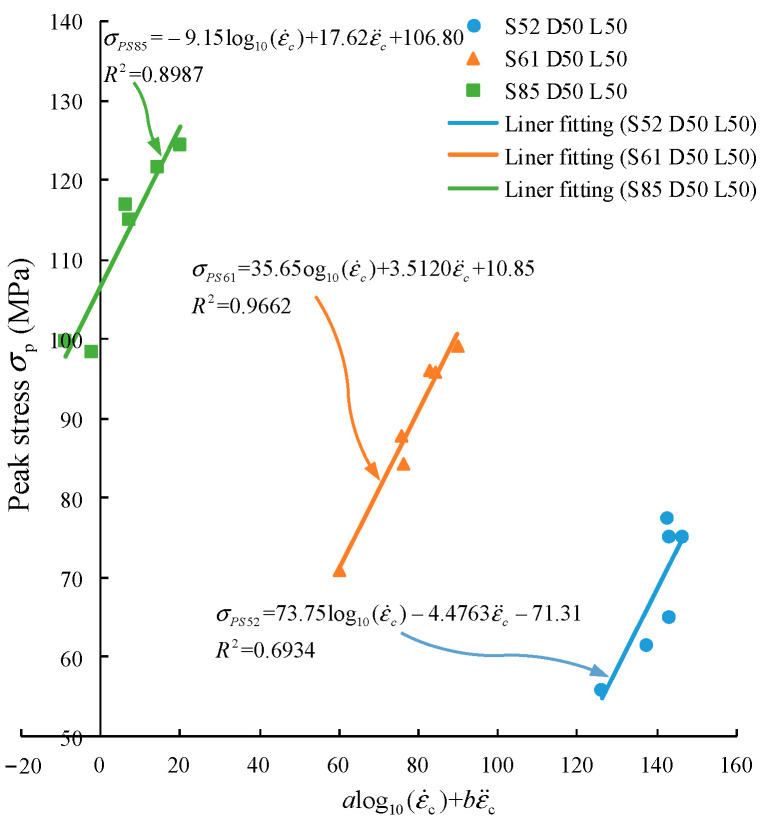
Comparison between the proposed model and the test results from Lee [47].

**Table 1 materials-15-00210-t001:** Micro-parameters of PFC model.

Material	Particle Radius (mm)	Density (kg·m^−3^)	Effective Modulus (GPa)	Normal-to-Shear Stiffness Ratio	Shear Strength (MPa)	Tensile Strength (MPa)
Steel bar	0.8–0.96	7800	300	3.0	1 × 10^100^	1 × 10^100^
Specimen	0.3–0.36	2500	63	2.0	135	135

## Data Availability

Not applicable.

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
