# Peer review of "Dynamic Response of Rock-like Materials Based on SHPB Pulse Waveform Characteristics"

_materials, 2021, doi:10.3390/ma15010210_

Round 1
Reviewer 1 Report
The paper: Dynamic response of rock-like materials based on SHPB pulse waveform characteristics, by authors: Bi Sun, Rui Chen, Yang Ping, Zhende Zhu, Nan Wu and Yanxin He presents the effects of different pulse types and pulse waveforms on the incident waveform and dynamic response characteristics of specimens based on particle flow code (PFC). This method is used for analysing the dynamic mechanical properties of materials. This research could be interesting for the engineers involved in material science, civil engineering and construction building.
In my opinion, the paper can be published in your journal after a MINOR revision.
Comments:
- The Abstract is short and well written. The main ideas of the manuscript are presented in the most effective and reasonable manner.
- The Introduction section is too short. Maybe, analyse more the previous results. The literature review can be improved.
- Figures 6, 7 and 13 are not visible enough. Please, try to fix this. This is only a suggestion.
- The analyses of the results can be improved. Try to add some aspects of the results which you have done already. You can have deeper analyses.
5.Emphasize the novelty of your research. What is the main contribution of your research?
Reviewer 2 Report
The manuscript is written and structured well. The results are described and discussed in details.
The contribution could be a useful addition and the content could be of the interest. In the opinion of this reviewer, the manuscript is recommended for publication subject to the following recommendations
1-How do the results of the numerical simulations as presented in this study are compared to real experiment investigation. The authors are advise to validate the model against other models or real data.
2-It is recommended to discuss how does the trapezoidal shape best fit the sine shape of the wave, and what its advantages over other alternatives.
3-It is seen from Section 3 and Section 4, that both the amplitude and the time rise have effect on the results. This section has been investigated by mean of parametric study, i.e, changing one parameter and watch the behavior while fixing the others.
It is recommended to show the behavior by changing the two parameters simultaneously at the same time. A 3D plot or any sort of scatter plots may show better indication considering the variation of the two parameters.
4-In the line of this study, it is worth referring to sensitivity analysis in: Composite Structures 133 (2015): pp.1177-1190. https://doi.org/10.1016/j.compstruct.2015.08.051
